# Predicting students' satisfaction with academic services at a multicultural engineering university in Bangladesh: A multiple regression analysis

Mwebesa Umar[1], Mahbub Hasan[2]*

**1** Office of the Registrar, Islamic University of Technology, Gazipur, Bangladesh, **2** Department of Technical and Vocational Education, Islamic University of Technology, Gazipur, Bangladesh

* m.hasan@iut-dhaka.edu

**Data Availability Statement:** All relevant data are within the manuscript and its Supporting Information files.

## Abstract

This empirical study sought to determine the levels of satisfaction among engineering students enrolled at a multicultural international institution in Bangladesh with a reputation for excellence. An assortment of first- and fourth-year undergraduate students participated in the study by completing an online survey. The study focused on selected determinants namely; support services (SS), campus life (CL), economic factors (EF) and University corporate image (CI). The researchers used a survey research design (SRD) to illuminate students' opinions and views. A multiple regression analysis (MRA) was used to regress opinions of 326 respondents who participated in the study. The disproportional stratified random sampling was used to determine the samples. The study was guided by two hypotheses. The study analyzed predictors of student satisfaction with academic services by employing standard multiple regression analysis. The findings showed that the four determinants SS, CL, EF and CI were statistically significant to predict students' satisfaction levels [$F(4,321) = 143.786$, $p < .001$]. It was empirically discovered that Support Services had the highest impact to the model [$\beta = .496$, $p < .05$] followed by university Corporate Image [$\beta = .365$, $p < .05$]. The findings showed that Campus Life and Economic Factors were not statistically significant ($p > .05$) in the model of predictors implying that they do not influence students' satisfaction levels on their academic career at the university. The researchers recommend that in order to maintain students' satisfaction levels on their academic experiences, universities should consolidate on support services provided to the students as well as improving their corporate image and world view.

## 1 Introduction

The trend of students' mobility and choices of academic career has significantly changed dimensions in institutions of higher learning across the globe. In this case, students are so keen about their levels of satisfaction in regard to academic and social services offered by

**Funding:** The author(s) received no specific funding for this work.

**Competing interests:** The authors have declared that no competing interests exist.

universities [1–3]. It has therefore become incumbent upon universities to rebrand and strategically profile themselves in order to draw students' attraction [3–5]. That is why the speedy increase of universities in individual countries has prompted managers of such institutions to ensure continued quality and improvement [6–8]. Generally, there has been visible dynamism in socio-economic, geo-politics, globalization and students' self-drive in pursuit of the diversity of academic career. More often, and upon making choices, potential students consider benefits of internationalization of education such as joint research projects, student exchange programmes, staff mobility, foreign students-based programmes and financial aid among others. In addition, there are other multiple factors that drive students' satisfaction in a university such as; prospects of employability, relevant curriculum, quality of teaching & learning, cost of studies & scholarships, superior infrastructure, students' welfare, conducive learning environment, campus facilities, quality of academic staff, academic advising, value for money, university ranking, social life campus safety among others [9–11]. Further still, multiple studies on students' continued mobility and satisfaction have revealed that students choose universities cognizant of the benefits of their career opportunities. The studies have shown that students consider different significant forms of situations and study environments that could increase their chances of; improved educational skills, academic and social life, networking relationships, gaining intercultural experiences, and potential employability [10, 12–20].

However, as universities endeavor to serve students assiduously, they face challenges of attracting quality students because of somewhat limited resources, Islamic University of Technology (IUT) inclusive. The need or policy requirement to recruit particular or a section of students into universities also comes with challenges. For example, apart from local students who get subjected to rigorous admission aptitude tests for purposes of screening the *best out of the best*, IUT admits foreign students based on *only* their previous academic qualifications without any other exclusive screening, the result of which most often, ends up with *not-highly-qualified* foreign students. Therefore, levels of satisfaction differ from students to students depending on home background and prior knowledge of university operations and privileges. Johnston explains that universities face challenges of what he calls attracting respectful students due to high competitions with other universities, probably in terms of outlook, corporate reputation and image, brand and prestige [5]. He adds that in most cases, it is hard to reach out to brilliant prospective applicants. It is assumed that the pursuit to quench university students' satisfaction depends on countless domains including university's conventional culture, vision, mission, core values, corporate image, accessibility, career prospects, market-driven curriculum, quality of faculty members and solid university management. As a corporate international university, IUT endeavors to satisfy its students given its peculiar nature of existence. This study delimited to four factors as predictors to student satisfaction at the multicultural environment of IUT: student services by the university, campus life, economic factors and university's corporate image.

## 2 Significance of the study

The main rationale of this empirical study was to predict students' levels of satisfaction with academic services offered at the Islamic University of Technology (IUT). The services were manifested as students' own experiences during their study period at IUT. Specifically, the researchers formed a set of variables that would be able to predict students' satisfaction of such academic experiences. The first rationale was to assess how support services offered by IUT predict students' satisfaction of their academic experiences. Secondly, to assess how campus life at IUT predicts students' satisfaction of their academic experiences. Thirdly, to assess how economic factors predict students' satisfaction of their academic experiences at IUT. Fourthly,

to assess how IUT's corporate image predicts students' satisfaction of their academic experiences. These factors were selected and adopted which were perceived to influence students' satisfaction levels of their academic experiences at IUT in comparison with other research studies in literature [1, 21–25]. The major contribution of the study was proposition towards improvement of university services to students to meet their career satisfaction. The researchers therefore believed that the study findings would provide solutions to improve students' academic experiences, review policies of support services, revise students' campus life, support them economically and also promote the university's corporate image. In addition, the study findings added value to the existing body of literature on students' satisfaction levels with the university services towards their academic career. Besides, the study findings formed a basis for further research on related areas of students' satisfaction.

## 3 Conceptual framework

The study focused on how best IUT students were primarily satisfied with the services rendered by the University in regard to their academic experiences, support services, campus life, economic factors and corporate image. The term satisfaction is, by literature, derived from the Latin word satis denoting 'enough' and–faction denoting facere meaning 'to do' or 'to make' [26]. The concept of satisfaction has been used widely in various disciplines most especially in marketing where the producer or supplier tries to satisfy consumer's or user's attitude in view of the consumer's belief and valuation about the product or services provided.

There are many theories that have been advanced to explain the concept of satisfaction. This study referred to the Expectancy disconfirmation theory (EDT) as its guiding theoretical orientation. The EDT was developed by Oliver, who opined that a user's satisfaction level is a consequence of the difference "between expected and perceived product performance, and expectations as predictions of future performance" [27]. He observed that satisfaction as a positive disconfirmation is realized when a product or service is quite better than expected. Contrary, when performance of a product or service is worse than expected there will be consumer's dissatisfaction denoted as negative disconfirmation.

On the basis of EDT theory, expectations of IUT students are based on their beliefs about the quality of engineering education the university provides. On the other hand, and in regard to the fulfilment of customer's needs, goals or desires, Hansemark and Albinson (2004) posit that satisfaction becomes an overall customer attitude towards a service provider like IUT or an emotional reaction to the difference between what customers anticipate and what they receive [28]. In the satisfaction literature studies, a number of theoretical approaches have been developed to expound the relationship between satisfaction (positive disconfirmation) and dissatisfaction (negative disconfirmation). In the case of this study, the researchers adopted Negativity Theory (NT) approach. Anderson, the proponent of this theory approach proposes that when expectations are strongly held, users will respond negatively to any disconfirmation [29]. Therefore, dissatisfaction will occur if perceived performance is less than expectations or if perceived performance surpasses expectations [30].

The study therefore sought to predict how well the four factors namely; support services, campus life, economic factors and corporate image measure students' satisfaction levels of their academic experiences at IUT and how such levels relate to Expectancy disconfirmation theory and theoretical approach of Negativity Theory.

## 4 Context of this study

The Islamic University of Technology (IUT) represents 57 member countries of OIC. By nature, any potential high school student wishing to join IUT from amongst OIC member

states would expect excellent and high-tech broad-based engineering education and services at campus. When students are applying to join IUT, they are provided with all the details pertaining to the university. Such particulars include; the history of IUT, faculties and departments, offered programmes, application and nomination procedures, fees and payment schedules and scholarship opportunities. The researchers considered the scope of the study in three perspectives; geographical scope, content scope and time scope. The geographical scope was delimited to IUT because of its nature of establishment as the first international university in the country. Other attractive details include; campus location and climate, ICT facilities, common facilities and services, medical services, library and e-resources, games and sports facilities, accommodation and feeding services, existing physical facilities, academic regulations and students' welfare associations. In addition to the above, students get acquainted to the university's vision, mission, core values and objectives of establishment.

The content scope, on the other hand, was delimited to five factors that attract satisfaction of students' livelihood: the academic experiences, support services, campus life, economic factors and IUT's corporate image. These factors were considered to be fundamental in the academic life of students in view of their academic studies, welfare, campus environment, economic survival and pride of the corporate image of their institution. In addition, the study considered time scope of the study. This is the period when majority of students expressed their dissatisfaction and cacophony arising out of mass failure of students in the final year examinations. Basing on the aforementioned profile of IUT, students get attracted to join expecting to receive real engineering education and services. However, in recent times, when they join and after spending a week or more often a semester or a year or two, some opt to drop out, change programmes/courses, express significant levels of dissatisfaction in particular academic areas and services though not pronounced amongst the majority.

## 5 Research questions and hypotheses

The purpose of this research was to examine the influence of several variables, such as campus life, financial position, and corporate image, on students' satisfaction in higher education. Additionally, the study sought to identify the most significant predictor linked to student satisfaction. The following research question was developed to direct the research process:

*How far do university support services, campus life, economic status, corporate image predict student satisfaction in higher education?*

In order to assess students' satisfaction with academic services, the researchers tested two alternative hypotheses;

H1: *The following set of determinants; [support services, campus life, economic factors, institutional corporate image] does not predict first years' and fourth years' satisfaction of their academic experiences in higher education institutions.*

H2: *Campus life (CL) is the best predictor of first year and fourth year students' satisfaction of their academic experiences at higher education institutions among other predictors; [support services, economic factors, institutional corporate image].*

## 6 Methods

### 6.1 Ethical considerations

This study followed the ethical criteria recommended by the Committee for Advanced Studies and Research of the university. Data from the survey was collected via Google form. Data collection began on September 15, 2022, and ended on January 15, 2023. Each participant gave informed consent before data collection. The consent form stated that participants' identities would be kept confidential. The Department of Research, Extension, Advisory, Services, and Publications (REASP) of the university issued ethical approval for this study (Ethical Clearance number: 23–001).

### 6.2 Research design and procedure

A survey research design (SRD) was considered as appropriate and convenient for the study. Creswell describes SRD as a quantitative research procedure in which a researcher administers a survey to either a sample or to entire population [31]. The SRD is designed to describe the attitudes, opinions, behaviors or characteristics of the population [32, 33]. The study also adopted a cross-sectional survey where data was collected in the same specific period of time where students expressed views about their satisfaction of IUT offered services in the academic year, 2018–2019. The study compared a cross-sectional survey of two categories of students; the first years and fourth years in view of their attitudes, beliefs, opinions and practices towards IUT service delivery.

### 6.3 Participants

The study population has been described in literature as a body of people or objects under consideration for statistical purposes [32]. In the academic year 2018–2019, the population of first years and fourth years was 1018 with 860 males and 158 females respectively. The first-year students included those studying postgraduate and diploma programmes. These two groups of students were taken into account in the study to forecast their levels of satisfaction in light of what they anticipated from IUT's academic services when they first enrolled and as they continued to attend as fourth-year students.

### 6.4 Sample

A sample is a representative collection of elements used to determine realities about the population [34]. This study relied on the tested Krejcie and Morgan's tables to guide in determining a sample size [35]. The Tables illustrate that a population of 1018 people attract a sample of 278, it was therefore noted that the sample collected (326 students) was over and above the expected minimum. This study adopted disproportional stratified random sampling where the population was divided into subgroups or strata and a sample selected from strata as single stratum. A google form survey questionnaire was distributed the participants of in each stratum randomly. The respondents were therefore grouped into first years and fourth years respectively and the data collection instrument was administered online.

### 6.5 Instrument

The researchers used a self-constructed survey questionnaire to collect data. A 5-likert scale (1- very dissatisfied. 2- dissatisfied. 3- neutral. 4- satisfied. 5- very satisfied) was used to capture respondents' satisfaction levels. The instrument (see S1 File) was divided into 6 sections; section A contained 5 items on biodata, section B; 9 items on academic experiences, section C; 10 items on support services, section D; 9 items on campus life, section E; 8 items on economic

factors and section F contained 10 on IUT corporate image. In order to ensure validity of the survey instrument, it was submitted to three experts in empirical research studies who vetted the items within their constructs by linking them with the study objectives and rationale. The outsourced experts then calculated the validity of the instrument using Content Validity Index (CVI) where items were rated on a four point scale of relevance, clarity, simplicity and ambiguity [36]. Based on the experts' opinions, some items were accepted as valid whereas others were discarded as either irrelevant, or unclear or ambiguous. After the calculations, the CVI was found above > 0.6 or 60% meeting the minimum requirement of a research tool.

The survey instrument's reliability was assured through consistency, dependability, and the ability to replicate research findings [37]. In order to ensure reliability of the instrument, the researchers conducted pilot study (pre-test) to ascertain degree of unreliability. The pre-test reliability was piloted online with 100 respondents at the university and tested using Cronbach's Alpha (α) which has been described as a coefficient appropriate for empirical research [38]. The calculation of Cronbach's Alpha coefficients was determined using SPSS version 25. All the 46 items used in measuring the instrument had a Cronbach's alpha coefficient of; academic experience (α = 0.860), support services (α = 0.826), campus life (α = 0.819), economic factors (α = 0.796) and IUT corporate image (α = 0.887) respectively. The coefficients indicated α above >0.7 for individual constructs; showing overall reliability of the entire survey questionnaire as 0.949, which confirmed that the instrument was very reliable. Using a Google form, the survey instrument containing 51 items, including biodata, was released online in expectation of obtaining the minimal sample size of 278 responders. It was distributed to all first- and fourth-year IUT students via the institution's website. Respondents were given four weeks to submit properly completed surveys. Using a printer, the researchers collected the surveys.

## 6.6 Data analysis

The study used two forms of data analysis: basic descriptive statistics of frequency and percentage, secondly, by using multiple regression analysis (MRA). The researchers used Students' Satisfaction in relation to their Academic Experience (AE) as one continuous dependent variable (DV) and four continuous independent variables (IV) i.e. support services (SS), Campus life (CL), Economic factors (EF) and IUT Corporate image (CI). Pallant explains that MRA gives two outcomes; how much of the variance in AE can be explained by SS, CL,EF & CI, secondly, it brings out the relative contribution of each of these IVs [39]. The researchers therefore conducted tests to determine the statistical significance of the results, in terms of the model output and the contribution of each IV. The researchers were also cognizant of the MRA assumptions and ensured they were not violated including sample size, multicollinearity and singularity, outliers, normality, linearity, homoscedasticity and independence of residuals.

**6.6.1 Demographic factors of the participants.** This study took a survey to assess satisfaction levels of first- and fourth-year students of IUT in regard to the academic services the university offered to students. The survey tool was posted online. The response rate was higher than expected with 326 respondents from the expected minimum of 278. The researchers analyzed data using MRA and descriptive statistics with the help of Statistical Package for Social Sciences (SPSS) version 25. The results from demographic data indicated that 326 students responded to the study survey with 197 (60.4) first years and 129 (39.6) fourth years. As a whole, 46 (14%) females and 280 (86%) males. This ably explained the fact that there was slim enrolment of female students at the university.

The findings showed that 254 (78%) were between 18–22 years of age, 61 (19%) were between 23–27 years and 11 (3%) were 28 years and above. This meant that IUT welcomes

youths and most notably, this is the age group that is naughty with online activities. The responses were characterized by the international students' outlook. The host country, Bangladesh registered most respondents with 290 (89%), followed by Cameroonians with 13 (4%), the remaining 7% responses came from Somalis, Comorians, Chadians, Ugandans, Nigerians, Gambians, Yemenis, Afghans and Pakistanis. The findings showed that students' responses cut across academic programmes with BSc. (CSE) as the highest with 99 (30%) respondents followed by BSc. (EEE) 81 (25%), BSc. (ME) 66 (20%), BSc. (CEE) 34 (11%), MSc. (Engg) 24 (7%), BBA 10 (4%) and BSc. TE 7 (3%). When the students were asked how they learnt about IUT, it was discovered that; (40.5%) learnt about IUT through its brochures, (31%) through IUT website, 3.3% through relatives, (13.5%) through Facebook, (7.7%) through friends and (4%) through continuing students.

## 7 Results

In order to assess students' satisfaction with academic services, the researchers tested two alternative hypotheses;

*H1: The following set of determinants; [support services, campus life, economic factors, institutional corporate image] does not predict first years' and fourth years' satisfaction of their academic experiences in higher education institutions.*

*H2: Campus life (CL) is the best predictor of first year and fourth year students' satisfaction of their academic experiences at higher education institutions among other predictors; [support services, economic factors, institutional corporate image].*

A multiple regression analysis output was used by the researchers to establish the real contribution of each independent variable. The standard regression approach was utilized to concurrently enter all factors into the model. Furthermore, the determination of the best predictor among the four independent variables was done by examination of the standardized coefficients of the model.

Table 1 above explains one model in which CL, EF, CL and SS have been included as predictors and AE as dependent variable. In the column labeled R is the value of the multiple correlation coefficient between the predictors and the outcome. Therefore, .801 (80.1%) is the simple correlation between AE and the predictors. The R2 in the next column is a measure of how much variability in the outcome is accounted for by the predictors. All the predictors together account for .642 (64.2%) and the remaining 35.8% is accounted for by other factors. The regression analysis was conducted to examine the relationship between the dependent variable, TotalAE, and the predictors TotalCI, TotalEF, TotalCL, and TotalSS. The overall model demonstrated statistical significance (F(4, 321) = 143.786, p < .001), indicating that the predictors collectively accounted for a significant amount of variability in TotalAE (see Table 2).

### 7.1 Testing hypothesis 1

Table 3 shows the results of the multiple regression analysis used to determine the association between the predictors TotalSS, TotalCL, TotalEF, and TotalCI and the dependent variable TotalAE. The combined predictors of TotalAE considerably influenced the total model, which was shown to be statistically significant (F(4, 321) = 143.786, p < .001). R2 = .64 indicates that the model explained 64% of the variance in TotalAE.

In terms of the individual predictor coefficients, TotalSS demonstrated a favorable and significant unstandardized coefficient (B = 0.447, p < .001), indicating that an increase in TotalSS of one unit is related with an increase in TotalAE of 0.447 units. TotalSS had a standardized coefficient (Beta) of 0.496, which represents a somewhat positive effect. Similar to TotalAE,

**Table 1. Model summary[b] of predictors.**

| Model | R | R Square | Adjusted R Square | Std. Error of the Estimate | Durbin-Watson |
|---|---|---|---|---|---|
| 1 | .801[a] | .642 | .637 | 3.77463 | 1.938 |

a. Predictors: (Constant), TotalCI, TotalEF, TotalCL, TotalSS

b. Dependent Variable: TotalAE

TotalCI showed a positive and significant unstandardized correlation (B = 0.303, p < .001), showing that an increase in TotalCI of one unit is correlated with an increase in TotalAE of 0.303 units. TotalCI's standardized coefficient (Beta) of 0.365 indicated a somewhat favorable effect.

TotalCL and TotalEF, on the other hand, exhibited non-significant associations with TotalAE. TotalCL exhibited a non-significant unstandardized coefficient (B = 0.047, p = .370), indicating that it did not influence TotalAE appreciably. Similarly, TotalEF exhibited an unstandardized coefficient that was not statistically significant (B = -0.025, p = .633), indicating that it had no impact on TotalAE. The model's constant term was 3.455 (B = 3.455, p = .001), indicating the predicted value of TotalAE when all predictors are zero.

The following are the correlations among the variables: TotalSS exhibited a significant positive correlation with TotalAE (r = .744, p < .001), TotalCL exhibited a positive but non-significant correlation with TotalAE (r = .596, p = .050), TotalEF exhibited a non-significant correlation with TotalAE (r = .458, p = .027), and TotalCI exhibited a significant positive correlation with TotalAE (r = .701, p < .001). All Variance Inflation Factors (VIFs) were within an acceptable range (VIF < 5), indicating that there were no problematic correlations between the predictor variables, as revealed by the collinearity diagnostics.

In summary, the first hypothesis H1 is partially accepted. University support services (SS) and corporate image (CI) significantly predict first years' and fourth years' satisfaction of their academic experiences in higher education institutions. However, campus life (CL) and economic factors (EF) do not predict first years' and fourth years' satisfaction of their academic experiences in higher education institutions.

## 7.2 Testing hypothesis 2

Overall, the findings imply that the significant variables explaining changes in TotalAE are TotalSS and TotalCI. In consideration of the (β) values and the magnitude of the t-statistics, the findings showed Support Services had the highest impact to the model [β = .496, p < .05] higher than IUT Corporate Image [β = .365, p < .05]. These findings can help guide measures to improve students' overall experiences and offer insightful information about the elements affecting student happiness in the context of higher education.

**Table 2. Analysis of variance ANOVA[a].**

| | Model | Sum of Squares | df | Mean Square | F | Sig. |
|---|---|---|---|---|---|---|
| 1 | Regression | 8194.554 | 4 | 2048.639 | 143.786 | .000[b] |
| | Residual | 4573.559 | 321 | 14.248 | | |
| | Total | 12768.113 | 325 | | | |

a. Dependent Variable: TotalAE

b. Predictors: (Constant), TotalCI, TotalEF, TotalCL, TotalSS

Table 3. **Results of linear regression analysis with simultaneous entry.**

|  |  |  | 95% CI | | | |
| --- | --- | --- | --- | --- | --- | --- |
| Variables | Beta | Std. Error | LL | UL | β | *p* |
| (Constant) | 3.455 | 1.040 | 1.409 | 5.501 |  | .001 |
| TotalSS | .447 | .048 | .353 | .542 | .496 | **.000** |
| TotalCL | .047 | .052 | -.056 | .150 | .043 | .370 |
| TotalEF | -.025 | .053 | -.130 | .079 | -.020 | .633 |
| TotalCI | .303 | .039 | .226 | .381 | .365 | **.000** |

Note.

*p < .05

In summary, hypothesis 2 is rejected. Campus life (CL) is not the best predictor, rather university support service (SS) is the best predictor of first year and fourth year students' satisfaction of their academic experiences at higher education institutions.

## 8 Discussion

The study took a research survey to assess first years' and fourth years' satisfaction levels of their academic experiences. The services were manifested as students' own experiences during their course of study period. Specifically, the study focused on for determinants namely, support services, campus life, economic factors and IUT corporate image. The researchers used a survey research design (SRD) to illuminate students' opinions and views. The study used a multiple regression analysis (MRA) to regress opinions of 326 respondents who participated in the study. The disproportional stratified random sampling was used to determine the samples using Krejcie & Morgan Tables [35].

The study was motivated by the following research question: "To what extent do student support services, campus life, economic status, and corporate image predict satisfaction in higher education?" The findings showed that indeed the four determinants SS, CL, EF and CI were statistically significant to predict students' satisfaction levels [$F_{(4,321)}$ = 143.786, p < .001]. It was empirically discovered that support services provided by the university (49.6%) and the corporate image (36.5%) of the Organization of Islamic Cooperation (OIC) influenced first year and fourth year students to join IUT to advance their academic career. The principle of MRA analysis inferred that for each unit increased on support services and improvement on IUT corporate image students' satisfaction levels of their academic experiences increased by 49.6% and 36.5% respectively [40].

The findings therefore implied that IUT should interest itself in the factors that were tested on students' academic experiences which appeared in the "survey instrument" for purposes of improving students' satisfaction levels. They included lecture rooms, quality of teachers, course content, teaching methods, academic advising, laboratory facilities, library services, university's automation system and timely feedback from teachers. This finding concurred with the findings of Stoltenberg (2011) who investigated the concept of student's satisfaction; a case of international students [21]. The study found that were satisfied with library facilities and general academic life. On the other hand, factors that were tested on Support Services included; campus accommodation services, food service in the cafeteria, career counseling & placement, medical services, communication between students and teachers, university administrative policies, friendliness of IUT staff in time of need, Wi-Fi free for students and support for improving English Language skills. In addition, factors that formed IUT Corporate Image in the 'survey instrument' included IUT as a subsidiary organ of OIC, IUT image in

your country, IUT image in Bangladesh, academic reputation of IUT, Islamic reputation of IUT, size of campus, suitable for career preparation, IUT's vision & mission, environment for international exposure and reputation for excellent staff.

The findings showed that Campus Life and Economic Factors were not statistically significant ($p > .05$) in the model of predictors. This was perceived to imply that whether the university focuses on them or not they do not influence students' satisfaction levels on their academic career at IUT. The factors tested on Campus Life in the 'survey instrument' included physical appearance of the campus, sports and recreational facilities, students' clubs & organizations, students' own government, campus security, campus social life, IUT physical location, students' discipline at campus and friendliness of students at campus.

Similarly, the factors tested on Economic Factors that never appeared statistically significant included financial aid, transport services for IUT students, tuition cost (fees), students' pocket allowance, distance from home, cost of leaving at campus, opportunity for casual jobs at campus, and personal survival at campus. This finding was perceived to mean that the financial aid (pocket allowances & tuition waiver) that students receive from the administration, the fees structure, the perceived cost of living in Bangladesh all such factors do not add to their satisfaction levels on their academic career at IUT. The researchers wondered why students were not satisfied with economic factors prevailing at campus; could it be that there are not casual job opportunities for them at campus unlike in other institutions [21]. Could it be because of high cost of living in the country? The issue would not be fees structure because IUT is one of the cheapest nonpublic universities in the country that charge a consolidated fees structure that includes accommodation, feeding, library fees, ICT fees among other additional fees. However, the understandable barriers could be harsh physical climate and communication barriers most especially for international students. Such factors make students become more disillusioned.

The findings of the study confirm the study's theoretical orientation. The IUT is an international university with students of diverse cultures, different ethnical backgrounds, habits, attitudes, goals, interests, educational experiences, expectations among other complex diversities. By nature, it is rather practically impossible to meet each student's academic expectations [41]. In the demographic presentation of findings, it was discovered that use of brochures (40.5%) was the most predominant source of information through which students get to know about IUT as a university compared to IUT website (31%), Facebook (13.5%), friends (7.7%), continuing students (4%) and through relatives (3.3%). This would imply that at the time of application, they refer to brochures/booklets that do not entail adequate information about the university in terms of infrastructure, programme curricular among others. This is quite contrary to other institutions where internet and media are the major formal channels of information [1].

## 9 Conclusions and recommendations

Basing on the rationale of the study and its findings, the researchers concluded that IUT students take interest to observe key factors that heighten their academic career and inconsequential factors that they can relinquish. Consequently, it can be concluded that IUT students do not take campus life and economic factors as key determinants of their academic career. This could be because of the nature, culture and attitude they encounter a long way. The following recommendations are made:

In order to maintain students' satisfaction levels on their academic experiences, the university should improve on support services and advance its corporate image. Support services can be improved through advanced academic practices, benchmarking and formal systems of

work. On the other hand, IUT corporate image can be improved through rigorous periodical adverts in print and audiovisual media.

The factors that were found insignificant ie *campus life* and *economic factors* should be a university priority and focus. Students' may not regard them as important to their academic career because of unknown reasons. It would be the responsibility of the stakeholders to intervene accordingly.

Further study to include survey on mass failure of students in same courses. Secondly a survey on students' satisfaction levels on the engineering education curricular in view of their prospective career. Thirdly a tracer survey on IUT Alumni's satisfaction levels of their training at IUT and their employability upon graduation.

## Supporting information

**S1 File. Survey instrument for assessing predictors of first year students' levels of satisfaction.**
(DOCX)

**S1 Data. Survey on student satisfaction.**
(SAV)

## Acknowledgments

We thank all participants. The authors are also grateful for the insightful comments suggested by the editor and the anonymous reviewers.

## Author Contributions

**Conceptualization:** Mwebesa Umar.

**Data curation:** Mwebesa Umar, Mahbub Hasan.

**Formal analysis:** Mwebesa Umar, Mahbub Hasan.

**Investigation:** Mwebesa Umar.

**Methodology:** Mwebesa Umar, Mahbub Hasan.

**Writing – original draft:** Mwebesa Umar.

**Writing – review & editing:** Mwebesa Umar, Mahbub Hasan.

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
