## [Decision Letter · Decision Letter 0]

14 Jan 2024

PONE-D-23-32796Predicting Students' Satisfaction with Academic Services at a Multicultural Engineering University in Bangladesh: A Multiple Regression AnalysisPLOS ONE

Dear Dr. Hasan,

Thank you for submitting your manuscript to PLOS ONE. After careful consideration, we feel that it has merit but does not fully meet PLOS ONE’s publication criteria as it currently stands. Therefore, we invite you to submit a revised version of the manuscript that addresses the points raised during the review process.

**Please see the reviewers comments and upgrade your manuscript. **

We look forward to receiving your revised manuscript.

Kind regards,

Qaisar Abbas, Ph.D

Academic Editor

PLOS ONE

3. For studies involving third-party data, we encourage authors to share any data specific to their analyses that they can legally distribute. PLOS recognizes, however, that authors may be using third-party data they do not have the rights to share. When third-party data cannot be publicly shared, authors must provide all information necessary for interested researchers to apply to gain access to the data. (https://journals.plos.org/plosone/s/data-availability#loc-acceptable-data-access-restrictions)

Reviewers' comments:

Reviewer's Responses to Questions

**Comments to the Author**

1. Is the manuscript technically sound, and do the data support the conclusions?

Reviewer #1: Yes

Reviewer #2: Yes

Reviewer #3: No

2. Has the statistical analysis been performed appropriately and rigorously? 

Reviewer #1: No

Reviewer #2: Yes

Reviewer #3: No

3. Have the authors made all data underlying the findings in their manuscript fully available?

Reviewer #1: Yes

Reviewer #2: Yes

Reviewer #3: No

4. Is the manuscript presented in an intelligible fashion and written in standard English?

Reviewer #1: Yes

Reviewer #2: Yes

Reviewer #3: No

5. Review Comments to the Author

Reviewer #1: Research objective and question is too much complex. Author should write in simple manner. Author should write research objectives as per the variables. Research questions should rewrite as per objectives. Each research questions should answer separately in the data analysis and result section. Author stated that Google form have been used for the data collections so how can author claimed that stratified random sampling have been used in the study ? So clarify the sample selection process. Sample size is appropriate. Author should rewrite research finding section and discussion section. Author should use statistical analysis techniques as per reframe questions.

Reviewer #2: A. Please restate and structure your abstract in 200 to 300 words based on the manuscript's important ideas. It must include the following sections: (1) introduction to the problem. (2) aims; (3) methods; (4) results; (5) conclusion; and (6) keywords with three parts: (a) discipline of the study; (b) concepts studied; and (c) methodologies and processes.

B. After the conclusions, incorporate this methodology' recommendations into your study.

Reviewer #3: 1) Please clearly highlight the hypotheses of this research and the methodology used to prove or disprove it.

2) The Abstract section does not highlight the novelty clearly.

3) The introduction section is vaguely written. Introduction section must be written on more qualitative way.

4) English language should be carefully checked and carefully check paper for language typos. There are few typo and grammatical errors in the paper.

5) Organization and formatting of the paper is poor.

6. PLOS authors have the option to publish the peer review history of their article (what does this mean?). If published, this will include your full peer review and any attached files.

Reviewer #1: **Yes: **GANPATSINH PATEL

Reviewer #2: **Yes: **Laurice Tolentino

Reviewer #3: **Yes: **Prof. Kitmo

---

## [Author Response · Author response to Decision Letter 0]

17 Jan 2024

Response to Reviewer #1 comments

We extend our gratitude to the esteemed reviewer for their valuable input on our manuscript.

Two hypotheses, denoted as H1 and H2, have been incorporated into Section 5, aligning with the research question. This addition aims to provide a more comprehensive understanding of the variables under investigation.

In the analysis section, the results of the hypothesis tests conducted through multiple regression analysis are presented separately. This ensures clarity in demonstrating whether the hypotheses are accepted or rejected.

The study employed a disproportional stratified random sampling approach. This involved dividing the population into distinct subgroups or strata, with a sample selected from each stratum as a single stratum.

A Google Form survey questionnaire was distributed randomly among participants within each stratum, ensuring a representative and diverse sample.

Response to Reviewer #2 comments

We thank the reviewer for this comment. We have now restated the abstract in 230 words, following the guidelines provided by the reviewer. 

Recommendations are now added after the conclusions.

Response to Reviewer #3 comments

We appreciate the reviewer's insightful comments.

1) Two hypotheses, denoted as H1 and H2, have been incorporated into Section 5, aligning with the research question. This addition aims to provide a more comprehensive understanding of the variables under investigation.

2) The abstract is now revised and highlights the novelty.

3) After careful restructuring of the whole article, the introduction now reflects credibility.

4) After a meticulous restructuring of the entire article, we are pleased to inform you that the introduction section reflects its credibility.

5) The organisation and formatting of the paper are now revised.

---

## [Editor Report · Decision Letter 1]

18 Apr 2024

PONE-D-23-32796R1Predicting Students' Satisfaction with Academic Services at a Multicultural Engineering University in Bangladesh: A Multiple Regression AnalysisPLOS ONE

Dear Dr. Hasan,

Thank you for submitting your manuscript to PLOS ONE. After careful consideration, we feel that it has merit but does not fully meet PLOS ONE’s publication criteria as it currently stands. Therefore, we invite you to submit a revised version of the manuscript that addresses the points raised during the review process.

**ACADEMIC EDITOR: Minor revisions required **Please ensure that your decision is justified on PLOS ONE’s publication criteria and not, for example, on novelty or perceived impact.

We look forward to receiving your revised manuscript.

Kind regards,

Qaisar Abbas, Ph.D

Academic Editor

PLOS ONE
---

## [Author Response · Author response to Decision Letter 1]

19 Apr 2024

We sincerely appreciate the valuable feedback provided by the academic editor and the reviewers on our manuscript. We have carefully addressed each reviewer's feedback, ensuring that all concerns were thoroughly considered and appropriately addressed.

Regarding the nature of our study, we wish to clarify that our research article is empirical, utilizing empirical data to support our findings and conclusions. We have ensured the accuracy and reliability of the data presented, adhering to established research methodologies and ethical standards.

Although we have previously addressed all points raised by the reviewers in our revision submission, we have included them again below for your convenience and reference. We believe that these revisions have significantly strengthened the manuscript, rendering it suitable for publication in PLOS ONE.

---

## [Decision Letter · Decision Letter 2]

14 Jun 2024

PONE-D-23-32796R2Predicting Students' Satisfaction with Academic Services at a Multicultural Engineering University in Bangladesh: A Multiple Regression AnalysisPLOS ONE

Dear Dr. Hasan,

Thank you for submitting your manuscript to PLOS ONE. After careful consideration, we feel that it has merit but does not fully meet PLOS ONE’s publication criteria as it currently stands. Therefore, we invite you to submit a revised version of the manuscript that addresses the points raised during the review process.

We look forward to receiving your revised manuscript.

Kind regards,

Sandro Vieira Soares, Ph.D.

Academic Editor

PLOS ONE

Additional Editor Comments:

Dear Author,

Please find below the reviewer's comments. In accordance with this review, my decision is Major Revision.

Best regards,

Primarily, I'd like to extend congratulations to the authors for their choice of article topic and the statistical methodology employed for analysis. I wish to highlight certain areas for improvement.

Given that this is a statistical study involving latent and observable variables, it would have been more appropriate to utilise Structural Equation Modelling (SEM), which offers advantages over regression.

It is essential to present all steps of the SEM for validating the hypotheses. Null hypotheses (Ho) should be provided for each alternative hypothesis (H1 and H2).

It's worth noting that the questionnaire employs an interval scale (ranging from 1 to 5). This detail should be included on text.

Reviewers' comments:

Reviewer's Responses to Questions

**Comments to the Author**

1. If the authors have adequately addressed your comments raised in a previous round of review and you feel that this manuscript is now acceptable for publication, you may indicate that here to bypass the “Comments to the Author” section, enter your conflict of interest statement in the “Confidential to Editor” section, and submit your "Accept" recommendation.

Reviewer #4: All comments have been addressed

2. Is the manuscript technically sound, and do the data support the conclusions?

Reviewer #4: Yes

3. Has the statistical analysis been performed appropriately and rigorously? 

Reviewer #4: No

4. Have the authors made all data underlying the findings in their manuscript fully available?

Reviewer #4: No

5. Is the manuscript presented in an intelligible fashion and written in standard English?

Reviewer #4: Yes

6. Review Comments to the Author

Reviewer #4: Primarily, I'd like to extend congratulations to the authors for their choice of article topic and the statistical methodology employed for analysis. I wish to highlight certain areas for improvement.

Given that this is a statistical study involving latent and observable variables, it would have been more appropriate to utilise Structural Equation Modelling (SEM), which offers advantages over regression.

It is essential to present all steps of the SEM for validating the hypotheses. Null hypotheses (Ho) should be provided for each alternative hypothesis (H1 and H2).

It's worth noting that the questionnaire employs an interval scale (ranging from 1 to 5). This detail should be included on text.

7. PLOS authors have the option to publish the peer review history of their article (what does this mean?). If published, this will include your full peer review and any attached files.

Reviewer #4: No

---

## [Author Response · Author response to Decision Letter 2]

14 Jun 2024

We thank the reviewer for their congratulations on our choice of article topic and the statistical methodology employed for our analysis. We appreciate the reviewer's feedback and are grateful for their constructive comments on areas for improvement.

We appreciate the reviewer's suggestion regarding Structural Equation Modelling (SEM) as a potentially advantageous approach for statistical analysis. However, in this particular study, our primary focus was to identify the key factors influencing students' satisfaction in an engineering university through regression analysis.

The collected data and study objectives were specifically tailored towards this goal, aiming to pinpoint the most significant predictor variable among the observed factors. While SEM indeed offers valuable insights by simultaneously considering latent and observable variables, its application might not align with the scope and objectives of our research.

Regression analysis provides a straightforward and widely accepted method for determining the relationship between independent and dependent variables, which suited our study's purpose effectively. Moreover, considering the complexity and resource demands associated with SEM, employing it without clear relevance to our research goals could potentially introduce unnecessary complexity and ambiguity to our findings. 

Here are a few examples of research studies conducted in the higher education field that utilized multiple regression analysis (MRA) without employing Structural Equation Modelling (SEM):

Koob, C., Schröpfer, K., Coenen, M., Kus, S., & Schmidt, N. (2021). Factors influencing study engagement during the COVID-19 pandemic: A cross-sectional study among health and social professions students. PLoS One, 16(7), e0255191.

Abdolrezapour, P., Jahanbakhsh Ganjeh, S., & Ghanbari, N. (2023). Self-efficacy and resilience as predictors of students’ academic motivation in online education. PLoS One, 18(5), e0285984. 

Tarman, B., & Kilinc, E. (2023). Predicting High School Students’ Global Civic Engagement: A Multiple Regression Analysis. The Journal of Social Studies Research, 47(1), 56-63. https://doi.org/10.1016/j.jssr.2022.02.001

Julián, M., & Bonavia, T. (2021). Understanding unethical behaviors at the university level: A multiple regression analysis. Ethics & Behavior, 31(4), 257-269.

While SEM is undoubtedly a powerful tool for certain types of analyses, its application is guided by the specific research questions and objectives at hand. In our case, regression analysis emerged as the most suitable approach for achieving our study's aims of identifying the primary predictor of students' satisfaction in an engineering university.

The information regarding the interval scale of the questionnaire (ranging from 1 to 5) has been included in the document. Specifically, this detail can be found on page 8, section 6.5, and further elaborated in Appendix A on page 20.

---

## [Editor Report · Decision Letter 3]

8 Aug 2024

Predicting Students' Satisfaction with Academic Services at a Multicultural Engineering University in Bangladesh: A Multiple Regression Analysis

PONE-D-23-32796R3

Dear Dr. Mahbub Hasan,

We’re pleased to inform you that your manuscript has been judged scientifically suitable for publication and will be formally accepted for publication once it meets all outstanding technical requirements.

Kind regards,

Sandro Vieira Soares, Ph.D.

Academic Editor

PLOS ONE

---

## [Editor Report · Acceptance letter]

28 Aug 2024

PONE-D-23-32796R3 

PLOS ONE

Dear Dr. Hasan, 

I'm pleased to inform you that your manuscript has been deemed suitable for publication in PLOS ONE. Congratulations! Your manuscript is now being handed over to our production team.

Kind regards, 

on behalf of

Dr. Sandro Vieira Soares 

Academic Editor

PLOS ONE